# Oral Squamous Cell Carcinoma and Concomitant Primary Tumors, What Do We Know? A Review of the Literature

**Mohammed Badwelan** [1] ![ID], **Hasan Muaddi** [2] ![ID], **Abeer Ahmed** [1] ![ID], **Kyungjun T. Lee** [1] **and Simon D. Tran** [1,*] ![ID]

1   Faculty of Dental Medicine and Oral Health Sciences, McGill University, Montreal, QC H3A 0G4, Canada
2   Department of Oral and Maxillofacial Surgery, King Khalid University, Abha 61421, Saudi Arabia
*   Correspondence: simon.tran@mcgill.ca

**Abstract:** Head and neck cancer is among the top ten cancers worldwide, with most lesions in the oral cavity. Oral squamous cell carcinoma (OSCC) accounts for more than 90% of all oral malignancies and is a significant public health concern. Patients with OSCC are at increased risk for developing concomitant tumors, especially in the oral cavity, due to widely genetically susceptible mucosa to carcinogenic factors. Based on fulfilling specific criteria, these concomitant tumors can be called second primary tumors (SPTs), which can be further categorized into metachronous and synchronous tumors. This research reviews the literature that investigated the concurrent OSCC with second or multiple primaries to improve understanding of the definition, classification guidelines, and its effect on cancer survival. It also highlights the current investigation methods, the variation of standard treatment approaches due to such a phenomenon, and preventive measures discussed in the literature.

**Keywords:** oral squamous cell carcinoma; second primary tumor; multiple primary cancers; synchronous; metachronous

## 1. Introduction

Head and neck squamous cell carcinoma (HNSCC) is ranked as the sixth most common cancer worldwide, with approximately 900,000 cases and more than 400,000 cases of incidence and mortality rate, respectively [1]. According to the International Agency for Research on Cancer (IARC), cancer of the oral epithelium is expected to impact around 400,000 people in 2020, with a mortality rate of 178,000 cases, ranking it 16th in incidence and mortality worldwide [2]. Oral squamous cell carcinomas (OSCC) account for more than 90% of all oral malignancies, with a higher prevalence among male than female. In addition, the geographical prevalence varies globally; For example, in Southeast Asian countries, OSCC is the most common cancer, whereas in Finland, it is the 16th most common cancer. The difference in global prevalence could be attributed to variations in exposure to carcinogenic risk factors such as tobacco (both smoking and smokeless forms) [3–5]. Compared to breast cancer, OSCC has a worse prognosis, with a 5-year survival rate of 50%, resulting in annual treatment costs of $2 billion [6]. Despite its rarity, a literature review reported variations for multiple primary malignancies ranging from 0.73 to 11.7% [7]. Furthermore, studies comparing the characteristics of OSCC to those of other second primary tumors are relatively rare. This could be attributed to the insufficiency of reporting protocols and understanding of considerations related to multiple primary cancers. Hence, knowledge regarding on SPT studies and their approach is essential; this further highlights the need to know the historical aspect of the classification and the protocol to allocate the topography of OSCC. Understanding such concepts while studying the characteristics of multiple primary carcinomas of OSCC will facilitate approaching affected cases and their studies.

## 2. Background and Fundamental Considerations

Studies for multiple primary tumors (MPTs) are exclusive of the retrospective type. Therefore, a standardized classification for MPTs, their definitions, patterns of growth, and as well as a systematic approach for localizing the tumor are critical to facilitating data retrieval due to the unstable and inexplicable metastasis of these malignant neoplasms [8].

The first criteria for MPTs was the one suggested by Billroth in 1889 [9], which considers that: 1. each tumor must have a different histological appearance; 2. the tumors must arise in different locations; and 3. each tumor must produce its own metastasis. The studies of MPTs remained as an interest until Warren and Gates published a comprehensive review in 1932 [10], proposing the following criteria for the diagnosis of MPTs: 1. each tumor must present a definite picture of malignancy; 2. each must be distinct; and 3. the possibility that one is a metastatic lesion from the other must be excluded. Warren and Gates' classification was found to be more practical and realistic by many workers in the field who adopted these criteria in selecting cases for their studies.

In 1933, Lund [11] advised on the difference in etiology and pathogenetic mechanisms between multiple cancers of the same organ and multiple cancers of different organs or tissues in his case series; this theory was expanded by the criteria of Moertel et al. (1977) [12], which considered MPTs as: 1. multifocal, where two distinct malignancies arise in the same organ or tissue; 2. systematic, arising in anatomically or functionally allied organs of the same system; 3. paired, arising in paired organs; and 4. random, occurring as a coincidental or accidental association in unrelated sites. Furthermore, Moertel et al. [5] classified multiple primary neoplasms into synchronous (neoplasms that appear at the same time or within six months) and metachronous (secondary neoplasms that develop after more than six months).

Notably, the terminology adopted in these studies is interchangeable; the first diagnosed primary tumor is also known as an "indexed tumor," which is indicated in some studies as the most extensive tumor area (largest nodule) in the surgical specimen or the tumor treated with curative intent. Hence the terms multiple primaries and second primaries can be interchangeable in the literature. However, multiple primaries will not indicate the chronological discovery of the tumors, while using the term second primary would be better reserved to indicate the tumor other than the indexed one [8]. On the other hand, using standardized coding to indicate the topographic location and histological type of both tumors (indexed or second primary) is crucial to conducting any study for multiple primary neoplasms.

With the previously mentioned classification approach, it became evident how important it is to adapt topographical and histological classification codes that provide a tool to locate cancer site and consequently facilitate the retrieval of data for research. Thus, the World Health Organization (WHO) publishes and regularly updates the International Classification of Diseases (ICD) [13], The ICD is a globally accepted tool to provide universal coding for diseases. It gives a format for medical codes to be used in the classification structure. This format makes it possible to record, analyze, interpret, and compare mortality and morbidity data collected in different countries or regions and at different times [14].

In contrast to more restrictive international coding, a report by Curtis and Ries in 2006 [15] used data based on Surveillance, Epidemiology, and End Results (SEER). This considers several factors, including cellular tumor invasion to the basement membrane (tumor behavior), histology, site of origin, laterality of paired organs (parotid gland and tests), and time since primary diagnosis to identify multiple primary cancers. In general, SEER considers all metachronous cancers (occurring two or more months after initial diagnosis) as separate primaries unless proven to be recurrent or metastatic by the medical record. Furthermore, the SEER registry has been adopted by North American cancer centers.

## 3. Second Primary Tumors and Field of Cancerization

The concept of field cancerization has been discussed in several published articles ever since it was first used by Slaughter et al. in 1953 [16]. It is essential to understand that

the oral cavity lining mucosa is a large genetically susceptible area to multiple carcinogens simultaneously [17]. Carcinogenic factors from tobacco, alcohol or viral infection will initiate the development of a stepwise multifocal cellular alteration in the oral mucosa ranging from hyperplasia to different degrees of dysplasia mainly due to genetic alteration in cellular oncogenes and tumor suppressor factors via multiple mutations and epigenetics modifications [18]. Furthermore, the invasiveness (in situ vs. invasive) and the abnormal tissue changes surrounding the carcinoma are notable factors to consider [19]. The lateral spread of cancer cells by salivary micro-metastasis, intraepithelial migration of the originally altered offspring cells, or local effects of the primary cancer foci to the surrounding tissue have all cast doubt on the theory that these multifocal cancerous coalesces [20]. As a result, the size of the region upon which field cancerization may exert its effect on normal tissue versus the possibility that SPTs may originate in an independent topographical area must be investigated further [17].

## 4. Carcinogenesis and DNA Damage

With significant advances in genetic understanding, many researchers believe that most solid tumors, including OSCC, are caused by genetic changes. Understanding the basic molecular mechanism of DNA alteration is vital when studying OSCC, from risk prevention to treatment [18,21,22]. A 2017 review by Jouher et al. [21], Clearly explains the genetic alterations in oral cancer. They discovered that cancer cells and solid tumors have altered chromosome numbers and misoriented chromosome attachment to microtubules as a result of chromosome segregation. The variation among human genomes is due to deletion and duplication collectively, or what is called copy number alteration. The last was reported to be the cause of converting primary OSCC to metastatic one. Chromosomal instability detected by fluorescence in situ hybridization in surgical specimens of non-small cell lung cancer is associated with poor survival. Other genetic processes include loss of heterozygosity, telomere stability, cell-cycle checkpoint regulation, and DNA damage repair, which have been linked to changes in tumor suppressor genes (APC and p53), variations in proto-oncogenes (Myc). In addition to the instability of genes controlling normal cellular processes (EIF3E and GSTM1), all of which contribute to the transformation of normal epithelial cells into cancer cells after passing through the dysplastic stage. Moreover, Jouher et al. found a correlation among genetic alteration and cancer-related behavioral and environmental risk factors, such as smoking and alcohol [21]. Despite previously proposed genetic mechanisms, Uddin et al. [23] reported that OSCC is a genetically nonfamilial disease caused by a sporadic DNA somatic mutation.

## 5. Behavioral Risk Factors for Oral Cancer

The overview of developing a concomitant primary cancer with OSCC can be seen in Figure 1. The phenomenon starts with the exposure of oral mucosa to carcinogenic risk factors, then the dysplastic epithelial changes until the development of primary oral squamous cell carcinoma (P-OSCC). Based on fulfilling the SPT classification criteria, including the time of discovering SPT, it would be considered synchronous or metachronous OSCC.

### 5.1. Smoking

Several studies have shown that smokers who have already been diagnosed with primary OSCC are more likely to get a second primary cancer. In a comprehensive case-control study, the odds ratios (ORs) for developing second aerodigestive tract cancers were 3.8 and 5.0 for cigarette, pipe and cigar smokers, respectively [24]. 20% of patients with second primary oral cancer smoked two packets per day [25], with an OR of 1.8 compared to the control [24]. Shiels et al. [26] supported this in their study that included 2967 cases of head and neck cancer from five cohort studies assessing the associations between smoking before a first cancer diagnosis and the risk of a second primary cancer. Shiels et al. [26] found that, compared to nonsmokers, those who smoke more than 20 cigarettes per day

was associated with increased second primary cancer risk among survivors of head and neck cancer (hazardous risk = 4.45; 95% confidence interval of 2.56 to 7.73).

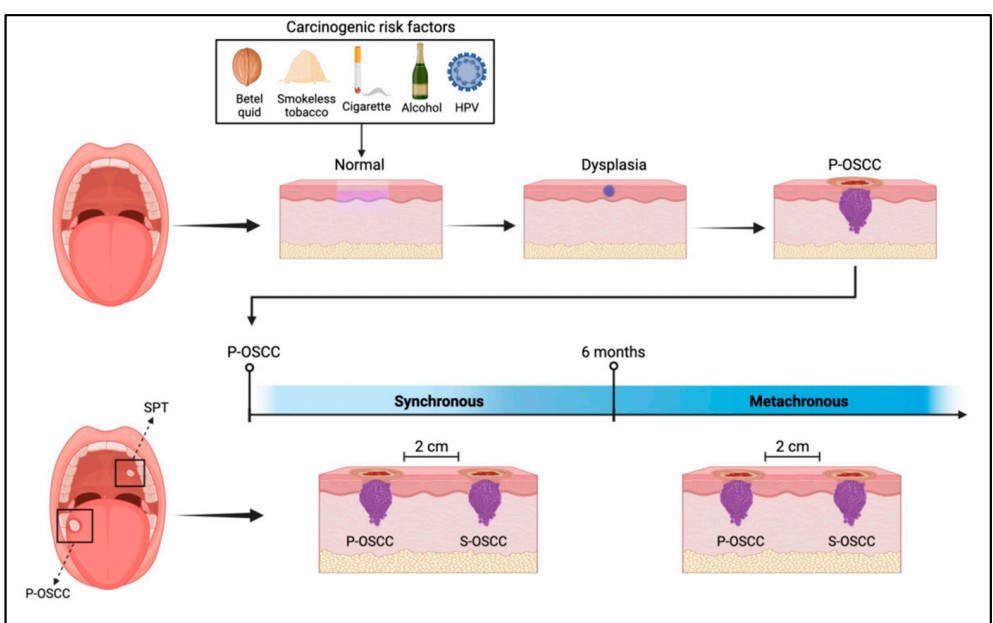

**Figure 1.** Shows the development of OSCC post-exposure to carcinogenic risk factors; If SPT was discovered before or after six months of P-OSCC development, it would be considered synchronous or metachronous, respectively.

### 5.2. Alcohol Consumption

Alcohol consumption is another major risk factor for oral cancer. It is suggested that the carcinogenic effect of alcohol is due to the intermediate metabolic products, resulting from the oxidization of ethanol into acetaldehyde by the alcohol dehydrogenase enzyme. The interaction between the resulted metabolic products and DNA will cause DNA mutation, and consequently cancer [4]. A variation in the amount of alcohol consumption and its subsequent categorization is found among studies. Leon et al. [27] divided patients into two categories based on their alcohol consumption: less than 50 g per day and 50 g or more per day. In comparison, Day et al. [24] divided patients into four categories based on their alcohol consumption: less than 5, 5–14, 15–29, and more than 30 drinks per week. The later study discovered that alcohol consumption is a major risk factor for developing a second primary cancer (SPC) after OSCC, with a 2.1 OR to develop another OSCC when drinking 15–29 drinks per week.

### 5.3. Both Smoking and Alcohol

Researchers have found that drinking alcohol and smoking together increased the risk of developing OSCC by 35 times. It is hypothesized that ethanol dissolves multiple carcinogens, making it easier for them to get into oral cells from tobacco [4]. In a matched case-control study, researchers analyzed the effect of the persistence of tobacco and alcohol use and the appearance of a second neoplasm in patients with head and neck cancer. Study results showed no statistical significance in the oral cavity and oropharynx cancer between cases and controls in relation to variables used in the matching, such as tumor stage, previous tobacco use, and alcohol consumption. However, the odd ratio for the appearance of the overall head and neck second neoplasm were 1.6 (CI: 0.9–2.5) and 11.2 (CI: 5.6–22.4) for moderate and high consumption, respectively [27].

A recent study retrospectively reviewed the database from Copenhagen oral squamous cell carcinoma registry. Of the 936 patients with primary OSCC treated with curative intent, 219 patients were found to have SPT. The study compared patients with and without SPC following a diagnosis of primary OSCC to examine the differences in baseline characteristics

between the two groups. Hazard ratios (HR) were used to express the association's potential. Tobacco smoking and alcohol consumption were among the variables analyzed using the multivariate Cox regression analysis. In contrast to the multivariate analysis, a significant association was found in the univariate analysis between the risk of SPC after treatment of OSCC and smoking and excessive alcohol consumption, respectively [28].

### 5.4. Betel Nut Chewing

Betel quid chewing (BQC) is a known risk factor for OSCC in the Southeast Asian population. In Taiwan, the percentage of OSCC among the betel quid chewer population varies from 59.7% to 82% [29]. BQC is associated with 10% of those premalignant lesions' malignant transformations, such as oral submucous fibrosis (OSF) [30]. BQC was found to accelerate oral submucous fibrosis into OSCC. In a retrospective study, of the 915 OSCC patients included, 25 of the 75 multiple primary oral cancer (MPOC) patients (33.3%) had CBQ-related OSF. In addition, patients with BQC habits and multiple primary oral carcinomas were of a younger age group than those with the other risk factors in the study ($p < 0.001$) [7]. The buccal mucosa was the most common primary occurrence site (35.9%) in MPOC cases, and almost all MPOC patients with buccal cancer had previously suffered from OSF (88.9%) [31].

### 5.5. Human Papilloma Virus

The role of the human papillomavirus (HPV) in OSCC is controversial due to the misclassification of OSCC and oropharyngeal squamous cell carcinoma (OPSCC). In addition, the HPV methodological approach for testing is inconsistent. This also impacted the precision of HPV as a risk factor in multiple carcinomas of the OSCC. According to several studies, HPV is the primary cause of OPSCC, with a better prognosis than non-HPV-induced OSCC [32]. In a multicenter Norwegian cohort study that included samples from 146 cases of OSCC (primary and secondary), histological p16 staining positivity was detected in 42% of the cases, with only two showing strong staining within the sample's cancer cells. Moreover, the results reflected the minimal etiological role of HPV in multiple OSCCs [33].

## 6. Diagnostic Modalities for Second Primary Tumors

Second primary tumors are often diagnosed in advanced stages, which leads to a low 5-year survival rate for affected patients. Therefore, early detection and screening of SPTs are vital to optimize disease-specific survival rates and reduce the burden of further treatment and dysfunction due to a better prognosis. Routine screening in the work-up and follow-up of patients with OSCC could detect more early-stage SPTs. The diagnosis of SPTs may impact the management and treatment of tumors [34].

### 6.1. Molecular Biology Techniques

Numerous molecular biology methods have been applied to pinpoint genetic anomalies linked to tumor growth and potential SPT predictors. p53 expression, p21, p73, and glutathione S-transferase polymorphisms were associated with the risk of SPT among patients with head and neck cancer. A quick and easy method for identifying head and neck cancer patients who are more likely to develop SPTs is immunohistochemistry labeling for p53 overexpression in tumor-distant epithelia; however, the accuracy of similar diagnostic techniques is still under investigation to assess their sensitivity and specificity [35,36].

### 6.2. Panendoscopy

With panendoscopy, the upper aerodigestive tract is inspected, including pharynx, larynx, upper trachea, and esophagus. It might also entail the biopsy or removal of any detected abnormal tissue. Using this method is the standard procedure for staging OSCC. As a result, it can be used to screen for simultaneous SPTs in cases of head and neck cancer. Panendoscopy in OSCC patients, including oro-, naso-, hypopharyngoscopy, laryngoscopy,

tracheoscopy, and esophagoscopy, could also result in faster treatment initiation and fewer treatment-related problems; however, it results in increased cost-effectiveness for the public health system [37]. Spoerl et al. [38] conducted a retrospective cohort study aiming to investigate the prevalence of synchronous OSCC within the upper aerodigestive tract (UAT) and specify patients' risk groups who would benefit from panendoscopy. Their study showed that, among the patients who had panendoscopy, 18 patients had a positive history of nicotine abuse and were found to have synchronous UAT tumors. The panendoscopy complication rate was 1.7% and mainly associated with dental trauma, except for one case of intraoperative esophageal rupture. In contrast to an earlier study conducted to detect SPT in the aerodigestive tract in OSCC patients without clinical signs of SPT, retrospective study results advised re-evaluating the need to panendoscopy in the cohort as only 0.8% of five synchronous SPT patients (1.9% of the sample included) were located in the area of panendoscopy and identified as having typical risk factors (alcohol and/or tobacco abuse) [39].

### 6.3. Positron Emission Tomography ($^{18}$F-FDG-PET/CT)

$^{18}$F-fluoro-deoxy-glucose positron emission tomography ($^{18}$F-FDG-PET/CT) is increasingly used for staging HNSCC and considerably impacts treatment decisions. $^{18}$F-FDG-PET/CT seems to be a useful diagnostic method for the early detection of SPT. Haerle and Strobel [40] discovered that $^{18}$F-FDG-PET/CT was superior to panendoscopy. They showed that panendoscopy detected fewer SPTs than $^{18}$F-FDG-PET/CT (4.5% versus 6.1%, respectively), but with more false positives. In a series of 589 patients with squamous cell carcinoma submitted to $^{18}$F-FDG-PET/CT, Strobel and Haerle [41] diagnosed 56 SPTs in 44 patients, 55% of them at the early clinical stage.

### 6.4. Narrow-Band Imaging Combined with Magnifying Endoscopy

Narrow band imaging (NBI) is a useful tool in diagnosing superficial squamous cell carcinoma (SCC) when combined with magnifying endoscopy in the oropharynx, hypopharynx, and oral cavity. NBI is an enhanced optical technique that enlarges the mucosal vasculature based on the correlation between the light wavelength and the depth of penetration. Hence, dysplastic squamous epithelial lesions and early SCC can be easily detected by their microvascular changes. A prospective study assessing NBI endoscopy for an early detection of secondary primaries after treatment of OSCC conducted by Giancarlo Tirelli et al. [42] showed that NBI endoscopy has an 89.5% sensitivity vs. 100% specificity, which could play a pivotal role in earlier stages and probably positively impact the surgical outcome and quality of life. Other studies reported that this diagnostic technique has a sensitivity of 94.7% and a specificity of 90.4% in detecting early-stage esophageal lesions, and enables a mean of minimally invasive treatment [43].

### 6.5. Alternative Diagnostic Methods

Other diagnostic techniques, such as Lugol chromoendoscopy (LCE), have recently gained popularity for detecting and monitoring early esophageal SCC. The technique uses Lugol's stain, which has produced encouraging outcomes. The targeted biopsy is made possible by isolating abnormal "mucosal islands" by Lugol's stain within otherwise normal esophageal tissue. This technique can locate optimal sites for a biopsy by locating suspected areas of mucosal lesions or premalignant change. Diagnostic accuracy is very high using Lugol chromoendoscopy [44]. The first systematic review was performed by Bugter and van de Ven [44] on the diagnostic yield of Lugol chromoendoscopy for esophageal SPTs in patients with HNSCC. According to their research, on average, 15% of primary HNSCC patients who underwent LCE had an esophageal SPT diagnosis. However, it is advised to investigate the application of such a technique in detecting second primary OSCC to assess its accuracy, cost-effectiveness, and practicality.

### 7. Management of Second Primary Tumors

Treatment of OSCC must follow well-established guidelines, such as those provided by the National Comprehensive Cancer Network (NCCN). These guidelines are based on the findings of well-controlled clinical trials, such as those conducted by the radiation oncology group. The management of patients with multiple OSCC or SPTs falls in the same area where patients are stratified according to their medical conditions and acceptance, institutional experience, primary tumor site, clinical stage, and other considerations [45].

#### 7.1. Surgery

Surgery remains the primary treatment for OSCC. The surgical management of neck nodes in patients with OSCC has been controversial, especially in the contralateral neck [46]. M. Garg et al. [47] proposed a strategy for managing the contralateral neck with ipsilateral second primary OSCC based on follow-up imaging, specifically positron emission tomography–computed tomography (PET-CT), in their 2019 review. If a hot area was seen in the PET-CT neck, neck dissection was recommended, while sentinel node biopsy (SNB) was recommended if the patient had a "cold" area in the SPT. However, patients with SPT are candidates for sentinel node biopsy if they have previously been treated neck cancer, and definitive treatment is required if the result of the SNB is positive.

Depending on when the second oral cancer was discovered, surgical resection would be more difficult with reconstructing two defects simultaneously in the case of multiple OSCC and the tissue required for reconstruction. Depending on cancer staging, defects may require only soft tissue reconstruction or soft tissue and bony reconstruction. Kao et al. [48] published a proposed algorithm for the reconstruction of separated defects that is based on retrospective observational data of patients treated for multiple OSCC, with the antero-lateral thigh flap serving as the main workhorse. The algorithm was based on assessing the disease prognosis, recipient vessel, and two defects' sizes, shapes, and types for those requiring only bone or soft tissue. Furthermore, these defects may require 1. one flap with consequent excision of the bridging tissue separating the two defects, creating a single defect, or 2. two flaps that can be further classified based on the reconstruction site and technique.

#### 7.2. Chemotherapy and Immunotherapy

The role of chemotherapy has been the focus of multiple clinical trials, whether adjuvant or neoadjuvant therapy [49]. In a recent propensity-matched analysis conducted by Lei Xiong et al. [50], the risk of second primary head and neck malignancy (SPHNM) in patients with locally advanced OSCC was assessed as an effect of chemotherapy against no chemotherapy. Adopting propensity score matching yields 10.7% and 22.1% for patients who received chemotherapy and did not, respectively, have a 10-year cumulative probability of developing SPHNM. Despite showing a 51% reduced risk of SPHNM using the regression model (0.49 adjusted subdistribution hazard ratio) within the chemotherapy group, there was no significant difference in disease-free survival between patients who developed SPHNM and those who did not. Moreover, according to a competing risk regression model based on a postmatch cohort conducted by Xinrong Li and Kaibo Guo [51], chemotherapy was negatively associated with the SPTs. The subgroup analysis was displayed by forest plots, demonstrating that patients with SCC, middle age (50–64 years old), male, good or moderate histological grade, unmarried status, and site of the tongue were more likely to benefit from treatment for reduced incidence of SPTs, which predominantly originated from the head and neck areas. Despite significant improvements in the incidence of SPTs, there was no statistically significant difference in survival between patients who received chemotherapy and those who did not.

On the other hand, immunotherapy for cancer treatment aims to amplify the immune system's ability to identify and destroy tumor cells. By changing their surface antigens, cancer cells can circumvent immune surveillance in the tumor microenvironment, preventing host lymphocytes from identifying and eliminating them. Tumors have the ability

to suppress the immune system by upregulating the production of ligands that can bind inhibitory T cell receptors. These ligands, also called immune checkpoints, work under normal physiological circumstances to block the progression of autoimmunity at a number of sites in the immune response [52].

Surface-expressed immune checkpoint inhibitors (ICIs), such as programmed cell death-1 (PD-1) and PD ligand-1 (PD-L1), are essential for activating negative regulatory pathways and avoiding the adaptive immunity [53]. Recently, a variety of monoclonal antibodies have been a major research area in treating various tumors. These ICIs have been shown to disrupt the transmission of inhibitory signals to T cells, therefore reprogramming adaptive immunity to assist in the clearance of cancer cells [54,55]. Expansion cohort performed by Chow et al. [55] to assess the antitumor activity of pembrolizumab, a highly selective monoclonal antibody that blocks the interaction between PD-1 and its ligands, in patients with recurrent and/or metastatic HNSCC. In patients with advanced HNSCC, the study revealed that the fixed-dose of 200 mg of pembrolizumab every 3 weeks was well tolerated and resulted in an overall response rate (ORR) with clinical significance and evidence of lasting responses. Furthermore, the response to pembrolizumab was much better in patients with HPV-related HNSCC than in those without HPV-related HNSCC, which has been linked to better survival rates.

A study by Wu et al. [56], on the other hand, revealed the variations in immunotherapy and chemotherapy sensitivity across distinct PD-L1 expression groups in HNSCC. Patients in the PD-L1 groups were shown to have a significantly higher likelihood of benefiting from ICI therapy, as well as a higher sensitivity to the four chemical therapeutics (olaparib, paclitaxel, docetaxel, and pazopanib).

### 7.3. Radiotherapy

Radiotherapy is a known cause of chronic inflammation of the oral mucosa; yet, the transformation of these mucosal changes into carcinoma remains controversial. Radiotherapy is considered a treatment modality for malignancies of the head and neck area, especially nasopharyngeal carcinoma (NPC). A study by Dai et al. [57] found that the second OSCC patient who survived NPC with a previous history of radiotherapy had a lower prognosis than those with sporadic OSCC. In contrast, Farhadieh and Otahal [58] discovered no significant increase in the development of SPT between patients who received radiotherapy versus those who did not. Intensity-modulated radiation therapy (IMRT) was developed to deliver a high radiation dose to the target area while avoiding vital structures. Ardenfors et al. [59] concluded that, when comparing the risk of developing SPT between IMRT and conformal radiotherapy, both radiotherapy techniques posed the same total level of risk, and IMRT only redistributed the risk in individual tissues.

Hashibe and Ritz [10] evaluated the impact of therapeutic radiation for oral cancer on the risk of SPTs, and the results of their study proved that radiotherapy had elevated risks of developing a second primary tumor. In another study, Song and Yang [60] investigated the clinicopathological characteristics and prognostic factors of second primary OSCC after radiotherapy for head and neck cancer patients and showed poor survival outcomes. Despite the previous controversies, many still consider radiotherapy to be the treatment of choice for radiosensitive tumors, and further studies investigating and differentiating the association from the causation of radiotherapy and SPTs are required.

## 8. Prevention of Second Primary Tumors

### 8.1. Modifying the Behavioral Risk Factors

Second primary tumors are prevalent; therefore, it is crucial to take prophylactic actions to reduce their incidence. El-Bayoumy K et al. [61] suggested an integrated approach to prevent OSCC that could be adopted to prevent the development of SPTs. The preventative method includes the cessation of the behavioral risk factors, such as smoking and nonsmoking tobacco, alcohol, and betel quid chewing. One-third of the SPT in HNSCC

primary tumors is attributed to persistent alcohol and tobacco consumption, and their termination must be a primary objective of index tumor treatment.

### 8.2. Chemoprevention

The use of chemoprevention (antioxidants) for SPTs was evaluated in various randomized clinical trials. Vitamin A and its isotypes are considered the most investigated chemopreventive agents. Isotretinoin (13-cis-retinoic acid) is a form of vitamin A used to prevent SPTs. Hong and Lippman [37] prospectively studied 103 patients who were disease-free after primary treatment for squamous cell cancers of the larynx, pharynx, or oral cavity. They found that daily administration of large isotretinoin doses successfully prevented SPTs in those who had been treated for HNSCC. However, it does not prevent the recurrence of the index tumor. Retinyl palmitate, also known as vitamin A palmitate, is the ester of retinol (vitamin A) and palmitic acid, whereas N-acetylcysteine is a medication derived from the amino acid L-cysteine. Van Zandwijk et al. [62] conducted a randomized intervention study on patients with head and neck cancer, most of whom had a smoking history. Patients with head and neck cancer who took retinyl palmitate and N-acetylcysteine supplements for two years showed no improvement in survival, event-free survival, or SPTs. β-carotene is a naturally occurring vitamin A precursor that acts as an antioxidant. Mayne et al. [63] investigated the effect of supplemental β-carotene on SPTs in a double-blind, placebo-controlled study. The results of the study however indicated that the decrease in SPTs incidence was not statistically significant. Furthermore, α-tocopherol is a type of vitamin E in humans and rodents that has been studied for its ability to treat SPTs. Bairati and Meyer [64] investigated whether antioxidants in vitamins, such as α-tocopherol and β-carotene supplementation, could reduce the incidence of SPTs in patients with head and neck cancer. They reported that supplementing with α-tocopherol had unanticipated negative effects on the likelihood of developing SPTs and cancer-free survival.

### 8.3. Vaccination against HPV

The vaccination against HPV is another approach to preventing OSCC, as the infection with the virus, specifically types 16 and 18, increases the predisposition to SPTs development [55,65]. Furthermore, increasing scientific evidence supports the national development of HPV vaccination programs, especially that of oropharyngeal carcinoma [66,67].

### 8.4. Further Research

Lastly, to aid the development of evidence-based follow-up advice after OSCC, future research should focus on risk stratification, the value of symptom-free detection of recurrences, and the active role that patients might play in determining their own follow-up regimen. Additionally, understanding the likelihood of a second primary, rate, and location, both by patients and the managing physician, will assist in shortening the time required to manage SPTs.

## 9. Characteristics of Studies Reporting OSCC and SPTs

In the literature, OSCC was considered the primary event, and the results of metachronous SPTs were reported as a secondary event (Table 1).

**Table 1.** Overview of screened literature reporting Second Primary Tumors (SPTs) in Primary Oral Squamous Cell Carcinoma (P-OSCC).

| Author | P-OSCC | Gender | | SPT | HNT | Non-HNT | SPT-AIT (Month) | 5 Years OS (%) |
|---|---|---|---|---|---|---|---|---|
| | | M | F | | | | | |
| Liu et al., 2013 [29] | 72 | 83% | 17% | 20 | 18 | 2 | 32 | 60.5 |
| Mochizuki et al., 2015 [22] | 1015 | 60.5% | 39.5% | 54 | 54 | N/A | 90 | N/A |
| Ko et al., 2016 [68] | 394 | 88.4% | 11.6% | 48 | 46 | 2 | 37 | 23 |
| Hu et al., 2018 [69] | 116 | 76.7% | 23.3% | 116 | 116 | N/A | N/A | 41 |
| Brands et al., 2019 [70] | 594 | 60% | 40% | 106 | 106 | N/A | N/A | 64 |
| Rogers et al., 2019 [71] | 347 | 61% | 39% | 29 | 29 | N/A | 52 | 39 |
| Kawasaki et al., 2020 [72] | 261 | 55.6% | 44.4% | 20 | 20 | N/A | N/A | 95 |
| Song et al., 2021 [60] | 48 | 66.7% | 33.3% | 48 | 48 | N/A | N/A | 39.4 |
| Petersen et al., 2022 [28] | 936 | 63.7% | 36.3% | 219 | 97 | 122 | 52.8 | 32.8 |

AIT: Average incidence time; HNT: Head and Neck Tumor; Non-HNT: Non-Head and neck tumor; OS: overall survival; M: Male; F: Female.

The overall survival rate (OS) of patients with only a single primary OSCC showed significantly better outcomes than those with multiple primary carcinomas or SPTs [22,68]. However, Kawasaki et al. [72] revealed that single primary carcinoma patients had an overall survival rate of 88.0% and 85.1% at 5 years and 10 years, while those with multiple primary carcinomas had a survival rate of 95.0% and 74.8% at those same time points. Discrepancies in OS may be linked to disparities in tumor stage, institutional qualities, and other variables, including host characteristics, comorbidities, and treatment approach.

Age is another major factor that can induce disparities in OS. On average, patients with SPTs were diagnosed at an older age than those with a single primary tumor at a median age between 63 and 68.3 years [22,71]. Kawasaki et al., 2020 [72], in contrast, found that the average age at first diagnosis for single primary oral cancer patients was 69.5 years, whereas those with MPTs were found to be 67.9 years old.

The most prevalent locations for SPTs after primary OSCC occurred in the head, neck, and lungs [28]. The oral cavity was the most common site for SPTs in the head and neck area; in the oral cavity, the tongue and gingiva were the most affected areas; however, tumors in the esophagus and liver were also recorded [68,69]. Mochizuki et al. [22] conducted a large retrospective study of 961 patients who had single-primary oral squamous cell carcinoma. During follow-up, they reported that 54 patients developed multiple primary carcinomas in the oral cavity, primarily in the gingiva and the buccal mucosa.

## 10. Conclusions

The medical-level improvement has increased the survival of OSCC patients. At the same time, OSCC survivors risk developing SPTs in the oral cavity, tongue, and gingiva. Although various screening methods exist to diagnose SPTs, the prognosis is not optimistic due to the different distribution of SPT sites and the different timing of their occurrence. Current diagnostic methods have limitations, including the inability to diagnose in the early stages, which can be overcome by potential molecular techniques that use the expression of genetic variants p53, p21, p73, and glutathione S-transferase polymorphisms. Additionally, continuing daily intake of antioxidant vitamins, such as Vitamin A and its isotypes, and quitting smoking and alcohol may lower the risk of developing SPTs. More focus on the prevention and treatment of SPTs is further needed. After treating primary OSCC, healthcare professionals must be aware of SPTs' risk, and patients should be advised regarding this situation. Future studies must focus on developing approaches to help with early screening of SPTs and their timely treatment.

**Author Contributions:** Conceptualization, M.B.; investigation and data collection, M.B. and H.M.; writing—original draft preparation, M.B.; visualization, M.B. and H.M.; writing—review and editing, M.B., H.M., A.A., K.T.L. and S.D.T.; supervision, S.D.T.; project administration, M.B. and S.D.T. All authors have read and agreed to the published version of the manuscript.

**Funding:** This research received no external funding.

**Conflicts of Interest:** The authors declare no conflict of interest.

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
