# Peer review of "Oral Squamous Cell Carcinoma and Concomitant Primary Tumors, What Do We Know? A Review of the Literature"

_curroncol, doi:10.3390/curroncol30040283_

Round 1

Reviewer 1 Report

This is a rare review that systematically introduces the concepts of primary and second primary lesions in OSCC and uses published data to prove the impact of second primary lesions on patient prognosis. It also puts forward in detail the technical means of monitoring and predicting the second primary lesion, emphasizing possible effective preventive measures. The purpose of this review is clear and distinctive, pointing out the direction of patient diagnosis and treatment and prognostic care and emphasizing the direction of future research on detection or supervision and prevention. The article is well written, it is recommended to accept!

Author Response

We appreciate the time you spent reviewing the manuscript. Thank you.

Reviewer 2 Report

Overview and general recommendation:

This article elaborates the occurrence and development of MPTs from the aspects of pathogenesis, risk factors, diagnostic criteria, treatments, and prevention of MPT, so as to help researchers and clinicians pay attention to OSCC and MPTs more comprehensively and develop systematic diagnosis and treatment approaches, thus improving the survival rate of patients. However, there are some small drawbacks need to be amended.

Major comments:

1.     Line 13 and line 46, what are the differences and correlations between the second primary tumors (SPTs) and the multiple primary tumors (MPTs)? Please, clarify it.

2.     Line 100, it is kind of confusing how multifocal cellular alterations can be induced from hyperplasia and dysplasia to malignancy or MPTs, the molecule mechanism needs to be further clarified.

3.     Line 108, there should be more references to support this view “most solid tumors, including OSCC, are caused by genetic changes”.

4.     Please check the full text carefully and revise the manuscript for any grammatical, stylistic, and proper noun errors.

5.     Please update the References.

Author Response

Thank you for reviewing the manuscript. We appreciate the time, effort and critical comments and advice that will make our manuscript presented more clearly.

Comment

Response

Correction location

Line 13 and line 46, what are the differences and correlations between the second primary tumors (SPTs) and the multiple primary tumors (MPTs)? Please, clarify it.

Thank you for spotting this overlapping terminology and advising us to clarify its uses. The clarification has been added.

Line 74 -77

Line 100, it is kind of confusing how multifocal cellular alterations can be induced from hyperplasia and dysplasia to malignancy or MPTs, the molecule mechanism needs to be further clarified.

Despite the available literature, the phenomenon has yet to be fully understood till date. Hence, multiple recommendations for delaying the time of SPT discovery to avoid misdiagnosis with field cancerization. We appreciate your comment, and the paragraph was written again to explain the process further and clarify the mechanism.

Line 104 - 108

3.     Line 108, there should be more references to support this view “most solid tumors, including OSCC, are caused by genetic changes”.

Reference was added

Line 113

Please check the full text carefully and revise the manuscript for any grammatical, stylistic, and proper noun errors.

Thank you for your advice. The Manuscript was checked, and references were updated.

Reviewer 3 Report

I am glad to have the opportunity to review this interesting paper. This manuscript provides a definition of second primary tumors (SPTs) in oral squamous cell carcinomas (OSCC), their possible mechanisms of occurrence, diagnosis, treatment and prevention.

Specific issues below:

1、  According to the definition of SPTs, diagnostic methods for distinguishing them from metastatic tumors are crucial. However, this manuscript lacks further elaboration on the specificity and sensitivity of the differential diagnostic methods.

2、  The manuscript lacks sufficient evidence on the role of chemotherapy in the treatment of SPTs in OSCC, which may hinder the reader's comprehensive understanding of this aspect of the disease. To address this gap, I suggest incorporating more relevant data and research on the use of chemotherapy for SPTs in OSCC.

3、  The disease management section could benefit from a more comprehensive description of available treatments, including immune checkpoint inhibitors, monoclonal antibodies, and other relevant therapies.

4、  More evidence should be cited to specifically demonstrate the role of HPV in the treatment and prognosis of SPTs of OSCC.

5、  If possible, it is recommended to include data on the age of patients with SPTs and the time interval between detection of the second tumour in Table 2.

Author Response

We highly appreciate your essential comments and critique. Kindly find the response to each valuable comment below.

Comment

Response

Correction location

1、  According to the definition of SPTs, diagnostic methods for distinguishing them from metastatic tumors are crucial. However, this manuscript lacks further elaboration on the specificity and sensitivity of the differential diagnostic methods.

The literature investigating the diagnostic methods of OSCC SPT is few and under continuous development. Our manuscript presented the available articles despite the lack of literature explaining their specificity and sensitivity in detecting OSCC. In addition, two more articles were added to improve the understanding of these methods.

Line 246 -251 Reference 39

2、  The manuscript lacks sufficient evidence on the role of chemotherapy in the treatment of SPTs in OSCC, which may hinder the reader's comprehensive understanding of this aspect of the disease. To address this gap, I suggest incorporating more relevant data and research on the use of chemotherapy for SPTs in OSCC.

More evidence was added explaining the role of Chemotherapy

Line 326-334

3、  The disease management section could benefit from a more comprehensive description of available treatments, including immune checkpoint inhibitors, monoclonal antibodies, and other relevant therapies.

The immunotherapy role was added to the chemotherapy section 7.2 in the manuscript.

Line 335-361

4、  More evidence should be cited to specifically demonstrate the role of HPV in the treatment and prognosis of SPTs of OSCC.

More references were cited

Reference numbers 33 line 215

And reference # 67 line 422

5、  If possible, it is recommended to include data on the age of patients with SPTs and the time interval between detection of the second tumour in Table 2.

Thank you for your valuable comment, despite the lack of consistency in reporting data related to SPT in OSCC. The schedule comprehensively presented the available data. The average incidence time was added; however, the collected studies did not present data on the sample’s age.

Reviewer 4 Report

This manuscript reviewed multiple primary squamous cell carcinomas occurring head and neck regionThe English used in this review is refined. the contents of this review are very important and has many implications for readers. But some modification was needed. According to circumstances, this manuscript should minor revise to our journal.

Minor points

1.    Reviewer requests deletion of Table 1 as described in the text. This contents in Table 1 are common code internationally, and we do not consider it necessary to include it.

2.    Use abbreviations appropriately.

●Multiple primary tumors (neoplasms)→MPTs

L53, L66, L74, and so on.

●Second primary tumors→SPTs

L271, L325, L335, L356, L349 and so on.

●oral squamous cell carcinoma→OSCC

L328

●head and neck squamous cell carcinoma→HNSCC

L349

3.    Please check how to describe the References.

L157, L159,  L353,  L358

4.    Please check grammar in English.

L167-169

5.    Please insert Reference.

L119

6.    Please correct as 18F-fluoro-deoxy-glucose positron emission tomography, or 18F-FDG-PET/CT

18F-fluoro-deoxy-glucose positron emission tomography, or 18F-FDG-PET/CT

7.    Please check first line in Table 2. Spelling in Bold. Is this OK?

Author Response

Thank you for your thorough comments and recommendations.

Comment

Response

1.    Reviewer requests deletion of Table 1 as described in the text. This contents in Table 1 are common code internationally, and we do not consider it necessary to include it.

2.    Use abbreviations appropriately.

●Multiple primary tumors (neoplasms)→MPTs

L53, L66, L74, and so on.

●Second primary tumors→SPTs

L271, L325, L335, L356, L349 and so on.

●oral squamous cell carcinoma→OSCC

L328

●head and neck squamous cell carcinoma→HNSCC

L349

3.    Please check how to describe the References.

L157, L159,  L353,  L358

4.    Please check grammar in English.

L167-169

5.    Please insert Reference.

L119

6.    Please correct as 18F-fluoro-deoxy-glucose positron emission tomography, or 18F-FDG-PET/CT

18F-fluoro-deoxy-glucose positron emission tomography, or 18F-FDG-PET/CT

7.    Please check first line in Table 2. Spelling in Bold. Is this OK?

Table 1 was deleted.

Abbreviations were corrected accordingly.

The reference description and Table 2 were corrected.

Other comments were addressed.

Round 2

Reviewer 2 Report

This version of manuscript has improved and can be accepted.

Reviewer 3 Report

The author has solved all my questions. I have no further questions.